# Survived the Glaciations, Will They Survive the Fish? Allochthonous Ichthyofauna and Alpine Endemic Newts: A Road Map for a Conservation Strategy

**DOI:** 10.3390/ani13050871

**Published:** 2023-02-27

**Authors:** Ilaria Bernabò, Mattia Iannella, Viviana Cittadino, Anna Corapi, Antonio Romano, Franco Andreone, Maurizio Biondi, Marcellino Gallo Splendore, Sandro Tripepi

**Affiliations:** 1Department of Biology, Ecology and Earth Science, University of Calabria, Via P. Bucci 4/B, I-87036 Rende, Italy; 2Department of Life, Health & Environmental Sciences, University of L’Aquila, Via Vetoio—Coppito, I-67100 L’Aquila, Italy; 3Consiglio Nazionale delle Ricerche—Istituto per la BioEconomia, Via dei Taurini 19, I-00100 Roma, Italy; 4Museo Regionale di Scienze Naturali, Via G. Giolitti 36, I-10123 Torino, Italy; 5Museo del Castagno, Via V. Emanuele, I-87013 Fagnano Castello, Italy

**Keywords:** amphibian, Urodela, conservation, *Ichthyosaura alpestris inexpectata*, Calabria, Italy, endemism, invasive fish, morphometrics

## Abstract

**Simple Summary:**

In Italy, there is a growing concern over the survival of an endemic subspecies of newt, *Ichthyosaura alpestris inexpectata*, commonly known as the Calabrian Alpine newt, due to the recent fish introduction and acclimatisation into three of the few localised lentic habitats where this glacial relict occurs. Results of a recent survey focused on this rare endemic taxon are presented. We provide updated information on the Calabrian Alpine newt distribution, adding two new localities, documenting local extinction at a few historical sites, providing a rough estimation of population size, and giving a description of breeding habitat features. The results of our pilot study will facilitate future research activities, conservation measures for the amphibian assemblage, and habitat management after fish introduction in the Natura 2000 site. We also pinpoint some actions useful to avoid the extinction of this remarkable taxon.

**Abstract:**

The Calabrian Alpine newt (*Ichthyosaura alpestris inexpectata*) is a glacial relict with small and extremely localised populations in the Catena Costiera (Calabria, Southern Italy) and is considered to be “Endangered” by the Italian IUCN assessment. Climate-induced habitat loss and recent fish introductions in three lakes of the Special Area of Conservation (SAC) Laghi di Fagnano threaten the subspecies’ survival in the core of its restricted range. Considering these challenges, understanding the distribution and abundance of this newt is crucial. We surveyed the spatially clustered wetlands in the SAC and neighbouring areas. First, we provide the updated distribution of this subspecies, highlighting fish-invaded and fishless sites historically known to host Calabrian Alpine newt populations and two new breeding sites that have been recently colonised. Then, we provide a rough estimate of the abundance, body size and body condition of breeding adults and habitat characteristics in fish-invaded and fishless ponds. We did not detect Calabrian Alpine newts at two historically known sites now invaded by fish. Our results indicate a reduction in occupied sites and small-size populations. These observations highlight the need for future strategies, such as fish removal, the creation of alternative breeding habitats and captive breeding, to preserve this endemic taxon.

## 1. Introduction

The biodiversity crisis is particularly dramatic in freshwater ecosystems [1,2]. This biodiversity crisis requires actions and conservation measures at multiple geographical scales, global, regional, and local, considering both conservation priorities and limited resources [3,4,5]. The most often used criteria to establish conservation priorities are taxon rarity and endemism (e.g., [6,7]). IUCN criteria consider the limited number of localities of a given taxon as a key factor for assigning high-risk categories [8], while other conservation projects (e.g., the EDGE of Existence programme) involve other evaluation frameworks, such as including endangered species that embody a substantial portion of distinct evolutionary heritage. The insular nature of lentic freshwater habitats has produced a high rate of endemism with small geographic ranges [2] (and reference therein).

Amphibians are nowadays subject to a plethora of possible threats and pressures, ranging from alien species’ introduction to pollution, emerging infectious diseases and habitat loss, which sometimes act in synergy and result in detrimental effects at both individual and community levels (e.g., [9,10,11,12,13,14,15]). Alien species in freshwater habitats are among the greatest threats to native biodiversity [9]. Indeed, for amphibians in general and newts in particular, fish introduction is considered a determining factor of pond-breeding populations’ decline and extinction [12,16,17,18,19,20]. Naturally, the impact of alien species is particularly alarming for native taxa with a restricted range.

*Ichthyosaura alpestris inexpectata* (Dubois & Breuil, 1983), known as the Calabrian Alpine newt, is an endemic taxon of Southern Italy considered to be a post-glacial relict with small and fragmented populations [21,22]. It represents a unicum of high biogeographical and conservation value, with populations exhibiting genetic variation likely originating from the Middle Pleistocene and possessing limited genetic diversity, which supports a long-standing isolation hypothesis [21]. Since the early 1980s, the newt has been known in five localities in the north of the Catena Costiera (a coastal chain representing the first mountain range of the Calabrian Apennines), in ponds and lakes of varying depths localised in clearings within beech forests at an altitude ranging from 850 to 1135 m a.s.l. [22,23,24,25,26,27]. All these sites currently fall within the Natura 2000 Network established under the EU Habitats Directive (92/43/EEC) (Table 1).

The Calabrian Alpine newt is assessed as “Endangered (EN)” in the Red List of Italian Vertebrates [28] because of the low number of occurrence localities, the continuous degradation and loss of the few sites due to the impact of human activity, and the recent introduction of allochthonous ichthyofauna in its core range. Since 2017, in fact, three fish species, *Cyprinus carpio*, *Gambusia holbrooki* and *Carassius* sp., have been observed in three newt breeding sites in the Special Area of Conservation (SAC) “IT9310060 Laghi di Fagnano”, originally devoid of fish fauna [23,24]. Moreover, other natural pressures, such as the transformation of habitats due to silting phenomena and drought, represent worrying critical issues [23,24], as was also found for another peninsular Italian amphibian species [5] (and references therein).

The aims of this study are threefold. First, we aim to provide an updated framework of the presence/distribution, status and threats of the Calabrian Alpine newt. Second, we aim to supply baseline ecological data for each inhabited site derived from recent systematic monitoring in the SAC Laghi di Fagnano. Third, we aim to provide the first demographic and population parameters. Specifically, we surveyed all spatially clustered wetlands in the protected area where the focal species was previously observed and the two recently discovered sites, thus giving the most updated information about *I. a. inexpectata* breeding habitats (e.g., habitat features and aquatic macroinvertebrate assemblages). Moreover, we collected data regarding abundance, body size and body condition in four sites (two new and two historical), expanding the information on morphometrics and sex ratio to obtain the first perceptions of demography.

## 2. Materials and Methods

### 2.1. Study Taxon

The Alpine newt *Ichthyosaura alpestris* (Laurenti, 1768) is widely distributed in most of western and central Europe and the Balkans [29]. In Italy, this urodele has a discontinuous distribution with three subspecies, two of which have a large distribution area and one of which is extremely localised. *Ichthyosaura a. alpestris* (Laurenti, 1768) is widespread in the Alpine area, *I. a. apuana* (Bonaparte, 1839) occurs in the Langhe hill system, Turin hills, Oltrepo’ Pavese (Northern Italy), the Tuscan-Emilian Apennines, the Maritime and Apuan Alps and a restricted and disjointed area within the Laga Mountains (Central Italy); conversely, *I. a. inexpectata* has a very limited range in Southern Italy [22,23,24,25,30,31,32,33] (Figure 1).

Adults of this medium-sized newt present an aquatic lifestyle in spring, enabling them to breed in ponds with aquatic and riparian vegetation, transparent waters and short periods of winter frost. Reproduction occurs mainly in May and June, and no autumn breeding events are known to date [31]. Paedomorphic phenotypes were observed in Lago Due Uomini and Laghicello [31]. Available data on ecological requirements, phenology and life history traits, and demographic information for *I. a. inexpectata* are scarce or outdated. Knowledge of feeding habits only refers to the population of Laghicello [34]. In addition, for this site, the rough estimation of a few hundred individuals of *I. a. inexpectata* (210–300) is the only one reported in the literature [27]. The same authors assumed that the total number of mature individuals in all populations (from Laghicello and Fagnano lakes) could be estimated between 840 and 1200 [27]. Much less is known about populations in Laghi di Fagnano, which have never been assessed.

### 2.2. Study Area and Site Description

We surveyed seven lentic habitats, ranging from temporary ponds to permanent lakes (Figure 2 and Table 2). Five sites fall within the Natura 2000 site “Laghi di Fagnano”, which protects a natural system of lakes and ponds with an elevation range from 1000 to 1100 m a.s.l. There are nine other amphibian species within this network of wetlands, in addition to *I. a. inexpectata* (Table 2). Due to this species diversity, the site was recognised as an Area of National Herpetological Relevance by the Societas Herpetologica Italica in 1998 (AREN—ITA022CAL002) [35].

The Special Area of Conservation covers a surface area of around 19 ha, and it extends mainly along the slopes of Monte Caloria (Fagnano Castello) along a sub-plateau belt in which high-grade metamorphic rocks outcrop. The local climate is strongly influenced by the warm and humid air masses from the Tyrrhenian Sea, which give rise to thick fog formations and frequent and abundant rainfall that help mitigate the long, dry summer [37]. The site falls within the lower supratemperate bioclimatic belt with upper subhumid ombrotype [38]. The ponds and lakes network are within beech forest formations (*Fagus sylvatica*), included in habitat 9210*, with a rich shrub and herb layer. The climatic conditions and soil characteristics allowed the formation of peat bogs consisting of a layer of sphagnum and a mosaic of different plants of humid and marshy environments linked with dynamic successions due to variations in water level [37]. A detailed description of the habitats and vegetation in the SAC is available at http://retenatura2000.regione.calabria.it/, (accessed on 12 January 2023).

Briefly, a higher and constant water depth of up to 2 m characterises “Lago Due Uomini” (hereinafter referred to as DU), which hosts submerged vegetation of *Potamogeton natans* and a massive presence of *Phragmites australis*. “Lago Trifoglietti” (TR) is a silting peat bog consisting of a thick moss layer of *Sphagnum auriculatum* with various species of *Carex rostrata*, *C. paniculata*, *Juncus effusus*, *Lysimachia vulgaris*, *Eupatorium cannabinum* and, in the most flooded areas, *P. natans* and *Eleocharis palustris*. Moreover, the lake holds a unique palynological archive that has yielded important information on the dynamics of Holocene vegetation and climate in southern Italy [36]. A spring flows into the lake from the north, whereas an outflow runs southward. To prevent dry-up during summer and complete infilling, the municipality of Fagnano Castello built a small earthen dam in 2000 that led to the formation of a small underlying pond [36] called “Trifoglietti inferiore” (TRIF). “Lago Fonnente” (FO) refers to an area including the main lake and an adjacent flooded meadow, which dries up in the dry season, and a pond, historically called “Fosso Armando” (FA) [39,40] which is adjacent but not connected with the other wetlands. FO shows well-preserved marsh vegetation with sedges and rushes and is characterised by *Typha latifolia* and *P. natans*. During the study period, the contiguous flooded meadow appeared almost completely dry in June, when numerous dead larvae of all three species of newts and tadpoles of *Hyla intermedia* were found. Therefore, the site was not visited further in subsequent samplings.

The new breeding sites of the Calabrian Alpine newt are constituted by two woodland ponds found in the localities of Piano di Zanche (PZ) and Pantano Lungo (PL) (municipality of Fagnano Castello) during a herpetological survey in 2015 and, after discovering, visited sporadically. The description of these habitats is given in Section 3.

### 2.3. Sampling Procedures

We sampled the five sites in the SAC Laghi di Fagnano and the two new ponds from April to October 2022, during and after the newts’ breeding season (May–July), once a month. PZ and PL were visited six times (except in May) due to a weather change (sudden thick fog) preventing sampling. All visits were conducted during daylight hours.

Newt sampling techniques differed across sites due to the intrinsic features of each. Sites TR and DU are the most extensive, and due to the morphology and presence of dense aquatic vegetation, these cannot be easily and completely surveyed. Therefore, the visual encounter survey method (VES) was used for the lakes TR and DU by walking across the entire wadable surface of the wetlands for at least 30 min, so as to cover, in that time, the entire pond perimeter (~14 m/min). While searching for newts, we also reported the detection of fish. For FO, FA, TRIF, PZ and PL sites, survey protocol required at least two observers to enter the wetland and capture animals using dip netting. At least three removal passes lasting 20 min were made during a single visit, removing the individuals captured at each pass to avoid multiple counts and applying the removal sampling scheme to estimate population abundance (see below).

All the captured newts were handled with latex gloves and temporarily kept in plastic containers filled with water from the pond and measured for body mass (to the nearest 0.01 g using digital scales), snout-vent length (SVL) and total length (with a ruler to the nearest 0.1 cm). A randomly selected subset of captured adults (n = 25) for PL in June was measured, so as to make the captured individuals’ number homogeneous. The sex was determined by typical secondary sexual characters. Recognition of newt species during larval stages was based on the comparative staging tables of Bernabò and Brunelli [41]. Paedomorphic newts were distinguished by gill slits and three pairs of external gills, and evident secondary sexual characteristics [39]. Once all measurements and data had been collected, newts were released into the capture pond.

At each sampling session, other amphibian species occurring and breeding in the investigated aquatic sites were also recorded.

### 2.4. Habitats’ Characteristics and Macroinvertebrates Assemblages

We described the features of the seven sampled aquatic habitats with different hydrological balances and subjected to different pressures (fish-containing and fishless ponds) by recording the size, the water permanence, the distances between each pond and the water’s physical–chemical characteristics.

To evaluate the biodiversity levels and trophic availabilities of the four breeding water bodies where adult newts currently occur, we investigated the structure of macroinvertebrate assemblages. Therefore, macroinvertebrate sampling was conducted in June and October, following a time–space-standardised quantitative procedure. In each monitoring site, we sampled three main mesohabitats in fordable zones (emergent and submerged aquatic vegetation, littoral and central sediments) using a Surber-type sampler (square opening: 25 cm × 25 cm) (Scubla s.r.l., Remanzacco UD, Italy) equipped with a 60 cm net of 250 μmesh and a plastic gathering glass. All samples were sieved through a 0.45 μm sieve (Endecotts Ltd., London, UK) and immediately stored in refrigerated containers with 80% ethanol. Once in the laboratory, macroinvertebrates were separated from sediments through a stereomicroscope. All organisms were counted and identified at the family level [42,43,44] to estimate the following parameters and indices: abundance (N), taxonomic richness (TR), Shannon–Wiener Index (H′) [45] and Margalef Index (D) [46].

Water physical–chemical parameters were assayed twice during the study, in the early summer and autumn, with three measurements along a transect at about 10–20 cm below the surface water. Specifically, temperature, conductivity, pH and total dissolved solids were determined with a multiparameter probe (Hanna Instruments, model HI991300). Dissolved O_2_ and relative saturation percentage were measured with a portable oxygen meter (Hanna Instruments, model HI98193).

### 2.5. Data Analysis

We analysed the dataset, collecting information on adults of the Calabrian Alpine newts captured in TRIF, FA, PZ and PL (Table 2). The sample size from the four populations in several months was too small since we captured only larvae and <3 individuals; therefore, for each population, we pooled the total number of captured males and females during the sampling period.

Operational sex ratio in each sampling was expressed as the proportion of males over the total adult number (males/(males + females)); deviations from equality were assessed via a two-tailed binomial test when more than ten newts were caught.

To assess the body condition (a proxy for physical condition, overall health and fitness [47]) of the four populations of Calabrian Alpine newts, we calculated the scaled mass index (SMI) [48]. This index provides a comparable measure across individuals of different sizes by rescaling body mass measurements to a standard length (here calculated for SVL) accounting for allometric growth [48]. We calculated SMI separately for males and females within each site, thus avoiding the scaling issue that results when comparing BCIs across groups known to differ in size [49]. SMI is computed as follows (with *Mi* and *Li* being the body mass and the linear body length, respectively, *bSMA* the scaling exponent estimated via the SMA regression of *M* on *L*, and *L*0 the mean length of the study population):$$\hat{M}i=Mi{\left[\frac{L0}{Li\;}\right]}^{bSMA}$$

We quantified the scaling exponent bSMA in R [50] by using ‘smatr’ package [51] from ln-transformed data.

The normality of morphometric data (body mass, SVL, total length and SMI) was analysed via the Shapiro–Wilk test and the homogeneity of variances was analysed via the Brown–Forsythe test. Data fit normal distributions and resulted homoscedastic; therefore, we used a one-way analysis of variance (ANOVA) followed by Tukey’s multiple comparisons test to determine whether there was a difference in body size and body condition between sex within populations and comparing females and males, respectively, among populations.

We also tested the effect of fish presence, macroinvertebrate diversity (in terms of Shannon index), site and month on the SMI using a linear model.

We used OriginPro, Version 2022b (OriginLab Corporation, Northampton, MA, USA) to produce plots.

We estimated newt abundance through the most recent field capture data to obtain updated information about TRIF, FA and PL (sites where sufficient numbers of adult newts were found). Considering that the sampling method we used can be ascribed to the constant probability of capture (each pass does not lower or increase the probability of capture of the subsequent pass, and the chance of sampling each individual is constant, both for their detectability and/or population stability) and the removal of captured individuals was performed at each pass, we used the whole capture dataset, setting the removal method = ‘Zippin’ [52,53,54]. The population size was estimated through the ‘FSA’ package [54] in R. We considered the highest population estimate in the month of greater detectability of adult individuals (present in the water during the breeding season).

## 3. Results

### 3.1. Current Distribution Framework and Habitats’ Characteristics

Based on our multi-year observations (1984–1992, 2005, 2015–2018) and during a sampling visit in June 2022, as well as from information gathered from locals and landowners, the occurrence of the Calabrian Alpine newt population reported in the SAC “Pantano della Giumenta” is seriously questionable, since the last observation refers to the early 1980s (Table 1 and Figure 2). In Laghicello, the presence of Alpine newts was verified during a single visit in April 2022; no fish were introduced, but the site appeared subjected to advanced natural silting phenomena (I. Bernabò pers. obs.).

During surveys conducted in 2022 in three study sites in the SAC Laghi di Fagnano (FO, TRIF and FA), Calabrian Alpine newts were observed at least once for each life stage, whereas in DU and TR we did not observe any (Table 2 and Figure 2b). A true absence was not proven, and we could not exclude its presence in the deep central areas of the lakes, which cannot be sampled except with methods other than those used. Individuals of the common carp (Cyprinus carpio) and of the Eastern mosquitofish (Gambusia holbrooki) were highly abundant in the lakes DU and TR, respectively; the presence of cyprinid fish (Carassius sp.) and G. holbrooki was also recorded in FO and TRIF, respectively, but at a low density (Table 2).

The two new records were confirmed to be breeding sites (Appendix A): PZ is a temporary pond, and the water comes almost exclusively from rainfall; conversely, PL is a large woodland puddle that presents a mostly stable hydroperiod thanks to a water flow from a spring. Both sites have significant tree cover and low O_2_ levels due to a high plant debris load; woods of Fagus sylvatica dominate the vegetation around the two wetlands, and no water vegetation (macrophyte or algae) is present. Water bodies’ environmental features (measured during the last field campaigns) are summarised in Table 2 and in Appendix A.

Considering the linear distances among clustered wetlands in the SAC of Laghi di Fagnano and the new sites, we found that the nearest was PL (835 m away from FA), whereas PZ occurs at a greater distance (1321 m from TRIF) (Figure 3).

About the amphibian community, among the ten species occurring in the SAC, only *Bombina pachypus* was not found. *Rana dalmatina* was the most common species (in 100% of the sites), followed by *Lissotriton italicus* and *Triturus carnifex*, and then *Pelophylax* sinkl *esculentus* (71%). *Salamandra salamandra*, *Rana italica*, *Hyla intermedia* and *Bufo bufo* were detected in 43% of the aquatic sites (Table 2). Syntopy of *I. a. inexpectata* with *L. italicus* occurred in five sites (FO, TRIF, FA, PL and PZ), and it also lived with *T. carnifex* in four ponds (FO, TRIF, FA and PZ).

Regarding aquatic macroinvertebrate assemblages, a total of 43 families, 15 orders, and 7 classes of invertebrates, cumulatively, were identified in the four study sites (TRIF, FA, PZ and PL). The taxonomic list of all organisms collected at each monitoring site is reported in Appendix A. All ponds in both sampling campaigns showed high diversity indices, indicating stable habitats for aquatic invertebrates. In particular, PL showed the highest values of diversity indices (TR: 25, H′: 3.056, D: 3.050), followed by FA (TR: 15, H′: 2.727, D: 2.261) and TRIF (TR: 17, H′: 2.555, D: 2.119), whereas PZ had the lowest ones (TR: 12, H′: 2.439, D: 1.591). Moreover, the macroinvertebrate community biodiversity increased in terms of number of taxa from June to October, due to the significant input of trophic resources associated with woody/leaf litter, with an overall autumnal increase of 15.7%.

### 3.2. Population parameters

In TRIF, we captured 53 adult Calabrian Alpine newts from May to September, with a maximum of 34 individuals captured in June (20 males and 14 females; estimated abundance 37 ± 3, 95% C.I. = 30–44) (Figure 4), whereas we observed numerous larvae from late summer to autumn, with many of these showing evidence of predation (i.e., tails and legs partially nipped). TRIF was the only site where we detected paedomorphic individuals (four males and three females); paedomorphic males and females had an average SVL (± SE) of 38.8 ± 0.4 mm and 41.8 ± 0.5 mm, respectively. In FA, 30 adult newts were captured from April to June, with a maximum of 15 captured adults in April (10 males and 5 females; estimated abundance 20 ± 8, 95% C.I. = 5–35) when the pond, characterised by a highly variable water level, presented its maximum water level (Figure 4). The pond was almost completely dry (level of about 20 cm) from the end of July until October, and only abundant larvae were observed. In PZ, we collected eight adults (five males and three females) of Calabrian Alpine newts in June, whereas in September, we found one female with a few larvae (Figure 4). The complete drying of the pond was detected in July and August. In PL, we detected adult Calabrian Alpine newts every month except May (see Materials and Methods); larvae were observed from the end of July to October, including overwintering larvae. Altogether, we captured 129 adults; the highest number of individuals was observed in June (33 males and 35 females) (Figure 4). The estimated maximum abundance of newts was 88 ± 14 (estimate ± SE; 95% C.I. = 62–115).

In individual samples with N ≥ 10, our results showed a male-biased sex ratio of 0.59 (*p* = 0.256) in TRIF. In FA, the proportion significantly differed from 0.5 both in April (0.67) and June (0.73) (all *p* < 0.05), whereas, in PL, a significant deviation from 50:50 toward males (0.65) occurred only in April (*p* < 0.05).

In all populations except for PZ (due to its small size), females were confirmed to have larger body sizes than males (F_7,116_ = 10.50, Tukey’s multiple comparisons test, *p* < 0.001) (Figure 5 and Appendix A). Overall, the average (±SD) SVL was 47 ± 3.5 mm in males and 53 ± 4.2 mm in females, whereas the average total length was 81 ± 5.4 mm and 95 ± 5.5 mm, respectively. Sex differed statistically in body mass within sites (F_7,116_ = 17.86, Tukey’s multiple comparisons test, *p* < 0.001). The mean body mass was 2.6 ± 0.5 g in males and 4.4 ± 1.1 g in females, showing no intra-sexual divergence between populations (F_7,116_ = 0.55, Tukey’s multiple comparisons test, *p* > 0.05). There were no differences among populations in females’ or males’ body mass, SVL or total length (Figure 5 and Appendix A). SMI differed statistically between sexes in each site, and males exhibited the lowest body condition index (Tukey’s multiple comparisons test, all *p* < 0.001); neither male nor female body conditions varied significantly among the four populations (Figure 5 and Appendix A). Moreover, we found no significant effect of fish presence, macroinvertebrate diversity, site or month on the SMI (R^2^ adj. = 0.073, *p* > 0.05 for all of the factors considered).

## 4. Discussion

### 4.1. Updated Distribution, Species Occurrence and Habitat Characteristics

In recent years, no effort has been directed towards surveying and monitoring the Calabrian Alpine newt. Our results update the information on the newt distribution, providing records of local disappearance from three historical sites and the discovery of two new breeding ponds.

The Calabrian Alpine newt was traditionally considered to occur in five localities (see Table 1): three in the SAC Laghi di Fagnano (FO, DU, TR), one in the locality Laghicello and one at the SAC “Pantano della Giumenta”. Their occurrence in this protected area, where a goldfish population (*Carassius*
*auratus*) has been stable for almost twenty years, has not been confirmed since 1984, and the population has to be considered extinct. In Laghicello, the endemic newt still occurs. In contrast, in two of the three historically known breeding sites in the SAC Laghi di Fagnano, DU and TR, the newts were not recorded after fish introduction in 2017 and 2019, respectively. However, past surveys (until 2018) applying VES, when the two lakes were fishless, allowed the detection of all newt species. Thus, VES may not be the most efficient way to detect newts in these two studied lakes and could lead to false absences. Furthermore, to avoid predation risk from fish, amphibians can increase shelter use and reduce activities [55,56], resulting in a lower probability of being detected and, likewise, reduced foraging activities and reproduction, two essential fitness components [57]. On the other hand, it is also well known that newts avoid fish-invaded habitats [58,59]. Another Alpine newt subspecies, the Bosnian Alpine newt *I. a. reiseri* (Werner, 1902) from the Prokoško Lake, was dramatically reduced or, more probably, disappeared from the lake following fish introductions [60].

In our case, if the event is not a local extinction, it is a demographic reduction since the detection probability is also a function of the abundance [61].

The progressive modification or loss (early desiccation and silting phenomena) of the few suitable habitats in which *I. a. inexpectata* breeds and the substantial threat posed by the introduction of invasive fish may lead to reproductive failure. This could result in the extinction of the endemic subspecies and those of other newts’ populations of community and conservation concern. Fish eradication would also bring benefits to the two other newt species that are syntopic with the Calabrian Alpine newt, i.e., *Triturus carnifex* (Annex II and IV) and *Lissotriton italicus* (Annex IV), which are protected under the Habitats Directive and categorised as NT and LC by the IUCN, respectively.

Knowledge of taxon distribution in a given area underpins most conservation efforts and planning [62]. In the context of the severe threat to the survival of the Calabrian Alpine newt, the discovery of two new occurrence localities (PZ and PL) is crucial for the newts’ persistence and highlights the importance of constant field monitoring to assess the species distribution at a local scale [63,64], which is a dynamic trait, determined by local extinction and new colonisation phenomena. These new breeding sites are small woodland ponds formed on natural depressions in well-preserved beech forests, away from the main roads and excessive tourism. Local landscape features include forest continuity and a network of ponds that allow the permeability of matrix habitats for dispersion. The linear distance and the narrow differences in altitude (about 200 m) separating the new ponds from the known ones of Laghi di Fagnano suggest that these sites may have been colonised through the dispersal of newts from the nearby locations. Indeed, the closest connection pathway may have been about 0.8 km from FO and FA toward the northeast for PL and about 1.3 km from TR and TRIF toward the southeast for PZ. Although fidelity to the breeding site is reported in the Alpine newt [65], movement capabilities over long distances, exceeding 1 km and up to 4 km, have also been reported [66,67]. Future studies should map all wetlands in the vicinity of the SAC and consider inter-pond movements, migrations and dispersal to disentangle newt population dynamics and better understand how to plan adequate conservation measures, such as pond creation. In addition, conducting investigations with new approaches, such as eDNA, could be helpful in detecting newts’ occurrence in suitable habitats.

At high altitudes (i.e., in the Alps), *I. alpestris* usually lives in oligotrophic sites where the water is very clear. In the Apennines, on the other hand, it inhabits low- and mid-altitude sites where water is frequently turbid, the maximum temperature is high and the environment is generally unpredictable [30]. We provided the first information on water parameters where the subspecies *I. a. inexpectata* breeds. The new ponds can be considered oligotrophic environments based on physical–chemical measurements and total hardness (the total amount of Ca and Mg ranged between 3.29 and 11.4 mg/L) [68]. Dissolved oxygen values are very low, suggesting a strong insaturation, most likely due to the scarce primary productivity and the high oxygen consumption triggered by a large amount of autochthonous and allochthonous plant detritus in the ponds. Here, we describe the first data on the potential feeding resources available for the Calabrian Alpine newt. Overall, the four studied ponds showed high diversity and a well-structured macroinvertebrate assemblage. Our biodiversity indices values are comparable to those calculated for other natural lakes and ponds [69,70,71] and higher than those measured in artificial, garden and urban ponds [72,73]. In October, the macroinvertebrate assemblages showed, in each pond, an increase in shredding invertebrates feeding on organic detritus derived from the riparian vegetation [74]. With regard to hydrological variation, there was a drop in the water level in October at all sites, particularly in FA and PZ. However, this lower volume of water did not negatively affect the macroinvertebrate assemblages and seemed adequate to sustain the ecological niches occupied by these organisms. Most invertebrates found in breeding ponds are involved in the diet, during the aquatic phase, of adults, paedomorphs and metamorphs of any subspecies of *I. alpestris* [34,75,76,77]. Typically, the Alpine newt shows generalist dietary habits and exhibits seasonal plasticity; it exploits temporary resources and differing prey taxa in relation to their cycles of availability and local abundance [34,75,76]. However, an individual shift towards a more specialist feeding strategy in artificial sites in response to increasing prey diversity has been reported [78]. In general, the abundance of invertebrate prey in the aquatic breeding sites allows newts not to leave these habitats to forage, a relevant factor for their conservation status [79]. Our preliminary analysis reported no significant correlation between the body condition index, prey species availability and fish presence. The contribution of the newts to the feeding ecology may be elucidated only with dietary studies that may also clarify the trophic strategy of the Calabrian Alpine newt. Newts act as predators in naturally fishless ponds [80,81]; therefore, it is also essential to investigate if the unavoidable trophic competition with introduced fish may become a decisive limiting factor. Indeed, knowledge of the trophic ecology of threatened and protected newts is crucial to evaluating possible threats [82], especially when dealing with the effects of fish introduction, and thus to adopting appropriate conservation policies [83].

### 4.2. Population Parameters

The ecology and demographic parameters of the Calabrian Alpine newt are scanty or outdated. Although long-term monitoring programmes are required for more robust estimations of trends in their distribution and abundance, our survey confirmed that the overall population of the Calabrian Alpine newt in its core range might be much smaller than hitherto assumed. Indeed, the only available study by Dubois and Ohler in 2009 [27] on the population of Laghicello roughly estimated counts of between 210 and 300 individuals. The authors supposed that the total number of mature individuals at the four historically known sites (Table 1) is probably much less than four times the estimated population size of Laghicello. Altogether, 90 adult newts were caught in the ponds within the SAC Laghi di Fagnano, whereas 138 adults were sampled at the two new sites. The highest number was found in permanent ponds. A few adults and larvae were found in PZ; this pond may completely dry out in the warm season and is replenished by late summer rainfall. The detection of larvae of the three newts at the end of September confirmed a late summer reproduction in this water body. However, its interannual hydroperiod is not well-studied, and it remains unclear what reproduction and metamorphosis success occurs at this pond. PL was confirmed to be a suitable breeding habitat of high conservation value for the Calabrian Alpine newt and other amphibians; here, the most abundant population is present.

Population monitoring and demographic estimates in the breeding sites of Laghi di Fagnano are pivotal for the persistence of the Calabrian Alpine newt after fish introduction. To ensure accurate population size estimates, we plan to adopt a monitoring scheme using aquatic funnel traps in lieu of dip-netting and a capture–mark–recapture approach.

Throughout the breeding season, we found statistically significant sex bias in caught newts towards males in FA (every sampling) and PL (in April). In each sampling event, males outnumbered females. The observation of distorted sex ratios in amphibians using counts or captures may reflect an actual ecological trait of the studied populations but may also be an artefact due to different capture probabilities between sexes [84], and references therein]. In other populations of Alpine newts, sex ratios have been reported to be either male- or female-biased [27,85,86,87]. According to Lanza and coauthors [31], both sexes remain in the water throughout the breeding season. However, males were more numerous than females at the breeding sites early in the year, and females outnumbering males in late spring and early summer represents a weaker trend in many newt species, including the Alpine newt [88]. Given our data, we can speculate that the sex ratio in the studied ponds during the breeding season is quite dynamic and is determined by locally specific factors. Further studies are needed on such dynamics, which may vary between different types of water bodies (e.g., temporary vs. permanent, temperature, food availability and habitat quality), influencing mating behaviour and the duration of the breeding season.

Body condition is a measure of an animal’s foraging success and physiological state [47,89,90]. In our estimations of body condition, remarkable intersexual differences were found within each population. Males’ and females’ body conditions of Calabrian Alpine newts did not vary significantly among the four study ponds, which were both with and without fish and had differences in their hydrological regime. Further long-term monitoring studies are required to determine the seasonal and inter-annual variation in SMI values for each population and the effects of fish presence on newts’ body condition.

Our results on body size are comparable with those available in the literature on *inexpectata* and other subspecies. Sexual size dimorphism was detected in our study populations, with females being longer and heavier than males, as previously reported in the literature [21,31]. 

### 4.3. Threats, Conservation Measures and Active Management

We suppose that within a few years, fish may cause the extirpation of the Calabrian Alpine newt and other native amphibian species from otherwise viable habitats. Meanwhile, the temporary ponds, although protected from the introduction of fish due to their hydroperiod, could likely disappear or become unsuitable habitats for amphibians in a changing climate. Omnivorous fish, such as carp, can cause an increase in water turbidity by damaging plant communities and accelerating eutrophication processes, thus compromising the habitat choice and reproduction of several amphibian species [91,92]. Predatory fish cause numerous adverse effects, such as direct predation on eggs and larvae, injuries due to predation attempts, resource competition, and changes in activity patterns and micro-habitat use [19,55,93,94,95,96]. Furthermore, both fish categories may damage native amphibian populations by pathogen transference [97].

We found paedomorphic Calabrian Alpine newts only in TRIF, and to date, paedomorphs have been reported to occur in two localities: Lago Due Uomini and Laghicello [31]. Facultative paedomorphosis is frequent in several Italian populations of *I. alpestris* [22,39,98,99,100] and is favoured in permanent aquatic habitats and where prey are abundant [77,99]. Evaluating the occurrence and incidence of alternative paedomorphic phenotypes should be of interest, considering the impacts of competition and predation. Indeed, it has been documented that introduced fish have determined the decline of metamorphs and the disappearance of paedomorphs from many newt sites in Europe [100]. A recent study showed several detrimental effects of the invasive *G. holbrooki* on paedomorphs of *Lissotriton graecus*, which exhibited avoidance behaviour and higher metamorphosis rates in the presence of fish [101].

The eradication or control of non-native invasive fish species in the lakes of Fagnano are undoubtedly unique and powerful conservation tools to improve the status of endemic and protected newts as well as the whole amphibian community. Given the different ecological and biological conditions of each water body, a targeted strategy combining several fish management techniques is necessary. Eradication programmes should include physical removal and chemical approaches [13,102,103,104] after a risk analysis to decide which strategy should be chosen and what the likelihood of success is. Considering the relatively high financial costs of eradication, the prioritisation of fish removal from ponds of higher suitability for newts could buffer the impact of fish presence [19]. Frog and salamander populations are expected to recover within a few years after fish removal from lakes [13,81,104,105,106]. The proximity of stable newt-breeding populations that can act as both sources and sinks may strongly favour the long-term persistence and newt recolonisation of lakes that have been returned to a fishless condition [12,19,107]. Therefore, in this emergency context, it is mandatory to create a network of small satellite artificial and semi-natural ponds, spatially connected to the historical sites, where to maintain newt populations until programmes to remove fish from lakes can be carried out. The identification of sites where to create suitable habitats should be realised through field surveys and the application of the most recent ecological modelling techniques to indicate the areas with major potential for retaining water, and also predicting changes in temperature and precipitation.

Regarding the newly discovered localities, both sites deserve regular interventions to guarantee water permanence and reduce silting phenomena by enlarging and deepening the ponds and removing excessive biomass accumulation. Furthermore, continuous surveillance is desirable to quickly notice any introduction of fish.

Finally, to deal with the ‘emergency’ situation, priority conservation action should be taken to preserve the Calabrian Alpine newt by implementing an *ex situ* conservation programme with research and conservation facilities in scientific institutions or zoos in Italy and Europe. The necessary *ex situ* initiatives will complement and support the *in situ* conservation of this highly endangered taxon, also considering that its disjunct distribution and its narrow range make this Alpine newt, by far, the most relevant biogeographic and conservation-related taxon compared to all of the other five *Ichthyosaura alpestris* subspecies [108]. Thus, the conservation of the Calabrian Alpine newt is not only important in terms of protecting taxonomic diversity but, being an evolutionarily significant unit, it assumes relevance for the purpose of better conserving the whole species’ genetic diversity [109].

## 5. Conclusions

Updated distribution information and ecological data are mandatory to start a specific management plan for an endemic taxon on the brink of extinction, which is deserving of urgent conservation actions. Our observations facilitate conservation and management activities in the SAC Laghi di Fagnano and the surrounding area. The radical ecological upheaval following fish introductions undermines the survival of the Calabrian Alpine newt and the other two syntopic pond-breeding newts of community and conservation interest. An effective long-term management strategy to restore disturbed habitats by removing fish cannot be further postponed, including pond creation to provide all the amphibian species with alternative habitats. Local biodiversity and ecological functions greatly benefit from the eradication or control of non-native fish. Intensive monitoring and a systematic collection of data to ascertain more in-depth key ecological requirements and population status over a long period are also essential.

## Figures and Tables

**Figure 1 animals-13-00871-f001:**
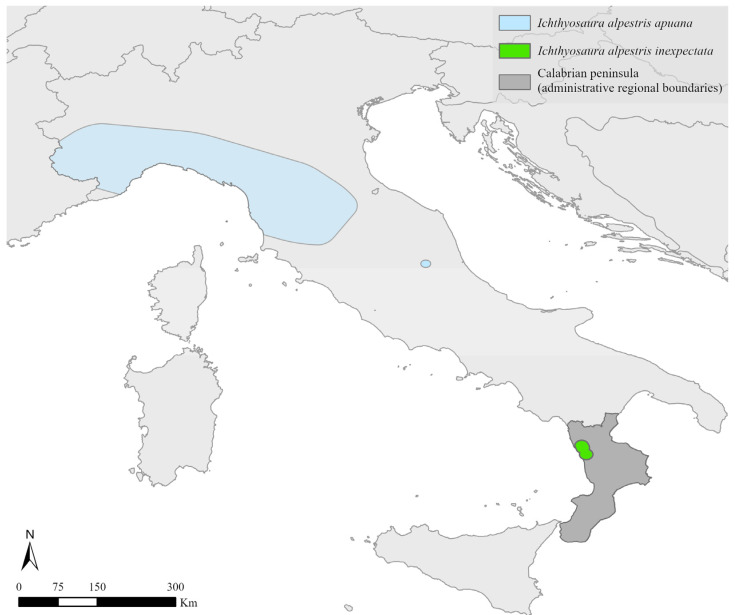
Italian ranges (minimum convex polygons based on [30,31]) of the two Italian endemic subspecies of the Alpine newt *Ichthyosaura alpestris*, namely, *I. a. apuana* (light blue) and *I. a. inexpectata* (green); in dark grey, the Calabria region, where our target subspecies (*I. a. inexpectata*) occurs.

**Figure 2 animals-13-00871-f002:**
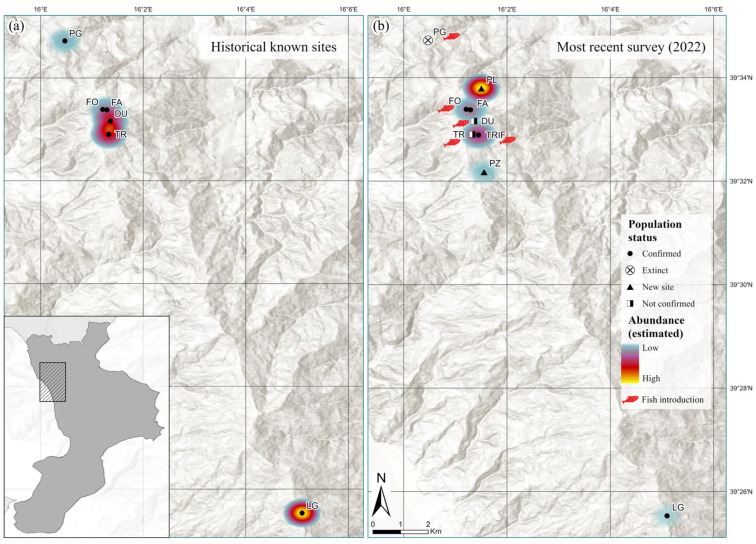
(**a**) Distribution and putative population sizes in the known localities of *Ichthyosaura alpestris inexpectata* based on literature data and historical observations; see Table 1 for details; (**b**) Results from recent surveys.

**Figure 3 animals-13-00871-f003:**
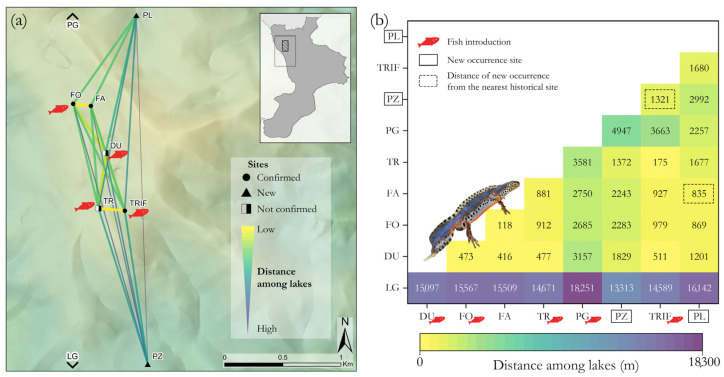
(**a**) Water bodies’ network within the SAC “Laghi di Fagnano” (Laghicello and Pantano della Giumenta sites are excluded from the view) and (**b**) the distances between each site (including all); the two new sites and the correspondingly shorter distances from which the Calabrian Alpine newt possibly colonised them are highlighted through boxes (full and dashed lines, respectively).

**Figure 4 animals-13-00871-f004:**
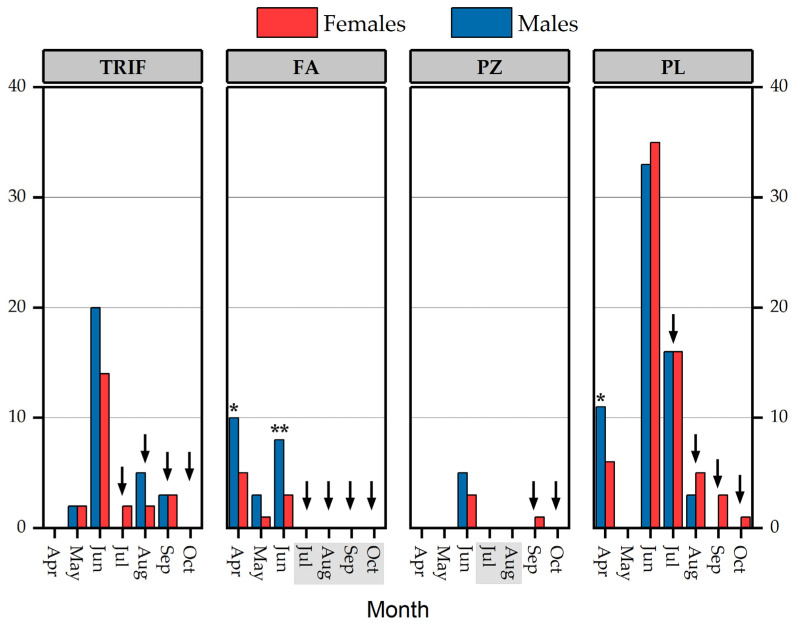
Captures of adults of *Ichthyosaura alpestris inexpectata* at four of the study sites during the sampling period. The months with no water are highlighted in grey. Asterisks mark a significant deviation from evenness, * *p* < 0.05, ** *p* < 0.01 (Fisher’s exact probability test); arrows indicate when larvae were found.

**Figure 5 animals-13-00871-f005:**
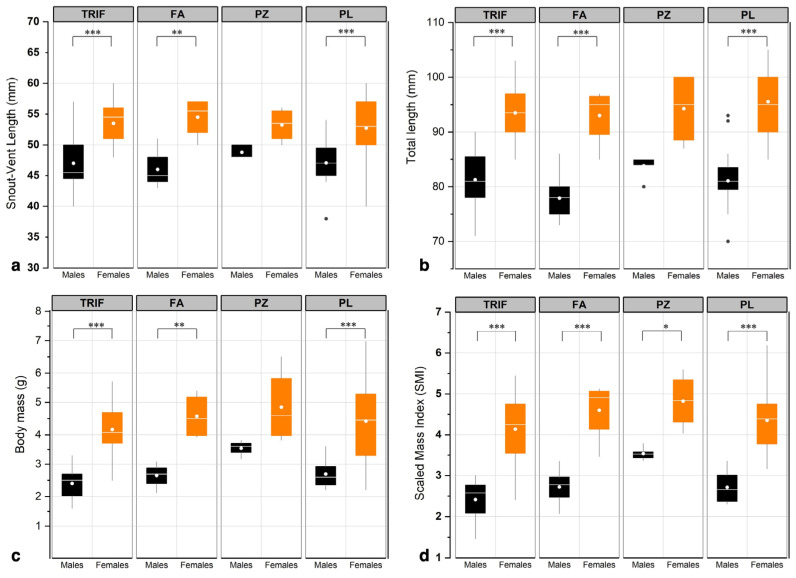
Boxplots of: SVL (**a**), total length (**b**), body mass (**c**), and SMI (**d**) for *I. a. inexpectata* males and females in the four sites. The horizontal white bars and circles represent median and mean values, respectively. The box is determined by the 25th and 75th percentiles, and the whiskers refer to the minimum and maximum values observed within data for each population. Outliers are shown as grey circles. One-way ANOVA with post hoc Tukey tests: * *p* < 0.05, ** *p* < 0.01; *** *p* < 0.001.

**Table 1 animals-13-00871-t001:** Summary of the data related to known sites. The last year of observation of *Ichthyosaura alpestris inexpectata* for each site is reported. Maximum number of adults captured or observed is based on information retrieved from the available literature or collected ante 2022 in the DiBEST database. Trifoglietti: TR = large pond Trifoglietti; TRIF = small pond Trifoglietti Inferiore. Fonnente: FO = Lake Fonnente; FA = Fosso Armando.

Site Code N2000	Municipality	Locality	Year of Last Observation	Maximum Number Captured/Observed (1983–2018)	References
IT9310058 Pantano della Giumenta	Malvito	Stagno C/Pantano dorato	1983	4	[24,25]
IT9310060 Laghi di Fagnano	Fagnano Castello	Trifoglietti (pond TR and pond TRIF)	2018 (TR) and 2022 (TRIF)	23	[22,23,24,25,26]
		Due Uomini	2018	21	
		Fonnente (lake FO and pond FA)	2022	25	[25,26]
IT9310061 Laghicello	San Benedetto Ullano	Laghicello	2022	69	[22,23,24,25,27]

**Table 2 animals-13-00871-t002:** Geographical locations, morphometric parameters and environmental characteristics measured in the study ponds. Site name: Fonnente (FO), Due Uomini (DU), Trifoglietti (TR), Trifoglietti inferiore (TRIF), Fosso Armando (FA), Piano di Zanche (PZ), Pantano Lungo (PL). The approximate area was calculated using Google Earth Pro satellite images (May 2022). Pond water depth (considering the maximum seasonal depths measured in April–May 2022), pond regime (P: permanent; T: temporary pond subject to drying out at least during a single survey), shading (i.e., the extent of shading of the site during the surveys) and presence or absence of fish are reported. Life stage: adults = A, larvae = L; overwintering larvae = OL; paedomorphs = P. Species abbreviations: Tla: *Typha latifolia*; Pna: *Potamogeton natans*; Pau: *Phragmites australis*; Cve: *Carex vesicaria*; Sau: *Sphagnum auriculatum*; Cro: *Carex rostrata*; Cpa: *C. paniculata*; Jef: *Juncus effusus*; Lvu: *Lysimachia vulgaris*; Eca: *Eupatorium cannabinum*; Epa: *Eleocharis palustris*; Lit: *Lissotriton italicus*; Tca: *Triturus carnifex*; Ssa: *Salamandra salamandra*; Hin: *Hyla intermedia*; Pes: *Pelophylax* sinkl *esculentus*; Rda: *Rana dalmatina*; Rit: *R. italica*; Bbu: *Bufo bufo*.

	Site Name						
	FO	DU	TR	TRIF	FA	PZ	PL
Lat. N Long. E	39°33′24″ 16°1′13″	39°33′9″ 16°1′22″	39°32′54″ 16°1′2″	39°32′53″ 16°1′27″	39°33′23″ 16°1′18″	39°32′11″ 16°1′34″	39°33′48″ 16°1′30″
Altitude (m a.s.l.)	1050	1077	1048	1045	1055	880	1010
Area (m^2^)	3561	20,679	10,800	520	827	742 ^1^	235 ^1^
Depth (m)	1	-	1.5 ^2^	0.9	1.2	0.7	1.1
Permanence	P/T ^3^	P	P	P	T	T	P
Shading (%)	20	20	30	80	20	90	90
Fish presence	*Carassius* sp.	*Cyprinus carpio*	*G. holbrooki*	*G. holbrooki*	-	-	-
Aquatic vegetation	Tla, Pna	Pna, Pau, Cve	Sau, Cro, Cpa, Jef, Lvu, Eca, Epa, Pna	Cve, Jef	Jef	absent	absent
*I. a. inexpectata*	L	-	-	A, L, P	A, L	A, L	A, L, OL
Other breeding species	Tca, Lit, Pes, Rda, Hin	Tca, Bbu, Pes, Rda	Hin, Pes, Rda	Lit, Tca, Ssa, Pes, Rda, Rit, Bbu	Lit, Tca, Ssa, Hin, Pes, Rda, Rit	Lit, Tca, Bbu, Rda	Lit, Ssa, Rda, Rit

^1^ Area calculated by measuring the maximum length and width and assuming an elliptical shape. ^2^ From [36]. ^3^ Temporary as referred to the adjacent flooded meadow.

## Data Availability

Not applicable.

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
