# Peer review of "Survived the Glaciations, Will They Survive the Fish? Allochthonous Ichthyofauna and Alpine Endemic Newts: A Road Map for a Conservation Strategy"

_animals, 2023, doi:10.3390/ani13050871_

Round 1

Reviewer 1 Report

I enjoyed reading the ms and congratulate the authors for this much needed study. Nevertheless I found it too long, it is more in the structure of a report than a scientific paper. I strongly recommend reducing it. There some minor aspects and corrections that should be considered.

Simple summary - I suggest removing mentions of Natura 2000 and site name since for the audience outside the EU this has little relevance.

L37 - CalabriaN

L42- rarefaction is not properly used.

L89 - aims 2 & 3 overlap partly.

Figure 1 - I suggest to use a map of the entire range of alpestris, not just in Italy

L117 - "deep" is not supported by the data presented. They are quite opportunistic and can even breed in shallow puddles.

L122 - I assume it is OUTdated

L261  "multi-year observations" when the inventory was conducted in 2022.

L271 - why were traps not used? They are quite efficient in capturing newts. See e.g. Arntzen, J. W., & Zuiderwijk, A. (2020). Sampling efficiency, bias and shyness in funnel trapping aquatic newts. Amphibia-Reptilia41(3), 413-420.

Table 2 - I suggest moving it to supplementary online materials

Figure 3 - same suggestion, supplementary online materials

L330 - while overall the assessment of macroinvertebrates is not properly justified (e.g. the presence of fish diminish the diversity of invertebrates), the statement that it increased is not clear - in what? Taxa diversity, number of individuals, biomass?

L367 - females were captured also during egg deposition and this affects body mass.

L413 - I don't see how long term monitoring benefits newts.

L441 - but nutrients were not measured during the study

L572 - while occasional drying of a pond can reduce reproductive success, it will eliminate fish and on long term is beneficial. I would not recommend this type of measure

Author Response

Response to Reviewer 1

I enjoyed reading the ms and congratulate the authors for this much needed study. Nevertheless I found it too long, it is more in the structure of a report than a scientific paper. I strongly recommend reducing it.

We thank you for the work on our manuscript and for the positive and constructive comments. We changed some parts of the manuscript accordingly to your comment.

There are some minor aspects and corrections that should be considered.

 Simple summary - I suggest removing mentions of Natura 2000 and site name since for the audience outside the EU this has little relevance.

Thank you for the suggestion, corrected.

L37 – CalabriaN

Corrected.

L42- rarefaction is not properly used.

Thank for the suggestion, corrected.

L89 - aims 2 & 3 overlap partly.

Corrected, we removed the term “demography” from the second aim.

Figure 1 - I suggest to use a map of the entire range of alpestris, not just in Italy

According to the other Reviewer suggestions and considering that the focus of the analysis of our manuscript is peninsular Italy, we preferred to remove the I. alpestris alpestris Italian range, and to keep and display the I. a. apuana and I. a. inexpectata ranges only.

L117 - "deep" is not supported by the data presented. They are quite opportunistic and can even breed in shallow puddles.

Corrected.

L122 - I assume it is OUTdated

Corrected, thank you for noticing it.

L261  "multi-year observations" when the inventory was conducted in 2022.

Thanks for this suggestion. We have rephrased the period specifying the years and periods in which we made the visits to this site.

L271 - why were traps not used? They are quite efficient in capturing newts. See e.g. Arntzen, J. W., & Zuiderwijk, A. (2020). Sampling efficiency, bias and shyness in funnel trapping aquatic newts. Amphibia-Reptilia41(3), 413-420.

We thank the reviewer for the suggestion on this point. We considered this year of surveys a pilot study to support conservation efforts and future monitoring. Thanks to appropriate funding, we plan to adopt a monitoring scheme using aquatic funnel traps to more effectively survey lentic habitats with different sizes, morphologies and features and to ensure accurate population size estimates in the next seasons. We recently tested the efficacy in Lake Due Uomini during an overnight sampling. Considering the environmental context, we are aware that this sampling technique is characterized by a high capture probability for all newt’s species and eliminates biases and habitat-damaging compared to active sampling methods.

Table 2 - I suggest moving it to supplementary online materials

Thanks for this suggestion, we prefer to maintain Table 2 (as it introduces to the reader, in a schematic fashion, all the relevant information of the study sites and the corresponding baseline information), but we have streamlined it by placing the part on the parameters in an additional table in the supplementary.

Figure 3 - same suggestion, supplementary online materials

Thanks for this suggestion, however, we prefer to maintain these two figures on the main text.

L330 - while overall the assessment of macroinvertebrates is not properly justified (e.g. the presence of fish diminish the diversity of invertebrates), the statement that it increased is not clear - in what? Taxa diversity, number of individuals, biomass?

Thanks for this suggestion. We have modified it to better clarify. 

L367 - females were captured also during egg deposition and this affects body mass.

We conducted the sampling during the peak of the breeding period. We captured mostly females with eggs. However, the analyses did not reveal any statistically significant difference in body mass between females with eggs and females without eggs; therefore, we pooled the female newts in the sites.

L413 - I don't see how long term monitoring benefits newts.

Corrected, we rephrased the sentence.

L441 - but nutrients were not measured during the study

The concentrations of nutrients (specifically phosphate and nitrate) were below the detection limit of the analytical methodology (ion chromatography, IRSA-CNR Manual 29/2003), so we have modified the text, as suggested, deleting all comments on nutrients.

L572 - while occasional drying of a pond can reduce reproductive success, it will eliminate fish and on long term is beneficial. I would not recommend this type of measure

We have modified it as suggested.

Reviewer 2 Report

Review report of the article “Survived the glaciations, will they survive the fish? Allochthonous ichthyofauna and endemic newts: a road map for a conservation strategy.”

By Ilaria BernaboÌ€, Mattia Iannella, Viviana Cittadino, Anna Corapi, Antonio Romano, Franco Andreone, Maurizio Biondi, Marcellino Gallo Splendore and Sandro Tripepi.

Summary: based on one field study and historical data, this paper gives precise and actual data about habitat description, distribution and abundance of an endemic Italian subspecies of alpine newt Ichthyosaura alpextris inexpectata. This study has been done in the context of the recent introduction of fishes in the ponds where this subspecies is known to live and breed. Based on these actualized data Authors give recommendations about conservation efforts and strategy in a context of emergency situation.

General comments

The manuscript clear, relevant for the field and presented in a well-structured manner.

Cited references are mostly recent publications (within the last 5 years) and relevant. It does not include an excessive number of self-citations.

The general methodology could benefit from the use of modern technics like the use of eDNA for the animal’s detection and habitat suitability models to reinforce the strength of the conclusions. Nevertheless, authors seem to be aware of it as they already mention some of the technics in the discussion as a perspective.

This study provides precise and important data on an ecological situation of primary importance. The conclusions of this study are essential for the pursuit of urgent conservation measures to protect this subspecies of Alpine newts.

Specific comments:

Line 55maybe it could be interesting to cite other programs of species’ prioritization according to conservation needs like EDGE program for example.

Lines 59-66Among other threats Authors should mention the risk of Bd introduction which could lead to a possible (and unpredictable) mass mortality in the ponds.

Lines 60-63: I would move this sentence as the first sentence of this paragraph and move the actual first sentence after it in order to first consider all the possible threats and then focus on invasive species.

Lines 69-70can we consider this subspecies as an Evolutionary Significant Unit (ESU) in need of conservation?

Lines 83-85Added to the fact that global change will increase the probability of extreme climatic events. Maybe Authors could add on sentence about it?

Lines 109-110: Authors could indicate the total number or recognized subspecies in the alpestris species?

Line 174: it could be useful the remind here the range of the breeding season.

Lines 175-176: what kind of weather change can prevent sampling?

Line 186: Please indicate if the operators wear gloves for this manipulation?

Line 239: Formula: it could be helpful to remind the reader the meaning of Mi, L0, Li and bSMA.

Line 337: What do you mean by “evidence of predation”? Can you describe these evidences?

Line 347: “I. a. inexpectata”: in line 345 you use the common name; I think you should be consistent with the use of scientific or common name in the manuscript.

Lines 372-373: have you tested the effect of presence or absence of introduced fished on the mean SMI of the newts? 

Line 475: general comment for the intro of the “Threats and conservation status” paragraph: it would be interesting to discuss the interest of focusing conservation effort not only at the species level but also at the subspecies level or even at the population or morphotype level that could be considered as Evolutionary Significant Unit who deserve conservation effort.

Line 519-520: does quantitative eDNA analysis could help to confirm the absence and reinforce population the density assessment?

Figure 1Depending on what authors want to illustrate on this map I would suggestion the following changes:

         1) if Authors want to present the localization of I. a. inexpecta among the other subspecies of Alpine newts, the map should be completed by extending I. a. alpestris to the North and completing the map in the Eastern part by adding I. a. montenegrina and a part of I. a. carpathica distribution too.

2) if Authors want to show only the two Italian "endemic" subspecies the nominal subspecies should be deleted from the map.

Figure 2: Please, indicate (a) on the left figure and (b) on the right one.

Figure 3: I would suggest to remove the LG site from the figure 3b as this site is clearly apart from the others (visible in figure 2) and do not appear in figure 3a.

Figure 4: Would it be possible to indicate on the graphs the months where larvae have been observed?

Table 1: 23 animals observed in Laghi di Fagnano: does this number refers to animals observed in 2018 and 2022?

Author Response

Response to Reviewer 2

Summary: based on one field study and historical data, this paper gives precise and actual data about habitat description, distribution and abundance of an endemic Italian subspecies of alpine newt Ichthyosaura alpextris inexpectata. This study has been done in the context of the recent introduction of fishes in the ponds where this subspecies is known to live and breed. Based on these actualized data Authors give recommendations about conservation efforts and strategy in a context of emergency situation.

 General comments

The manuscript clear, relevant for the field and presented in a well-structured manner.

Cited references are mostly recent publications (within the last 5 years) and relevant. It does not include an excessive number of self-citations.

The general methodology could benefit from the use of modern technics like the use of eDNA for the animal’s detection and habitat suitability models to reinforce the strength of the conclusions. Nevertheless, authors seem to be aware of it as they already mention some of the technics in the discussion as a perspective.

This study provides precise and important data on an ecological situation of primary importance. The conclusions of this study are essential for the pursuit of urgent conservation measures to protect this subspecies of Alpine newts.

 We thank you for the positive feedback and the useful comments on the manuscript, which we addressed in the new version. Below, we report responses to specific comments, which were all addressed and solved.

Specific comments:

Line 55: maybe it could be interesting to cite other programs of species’ prioritization according to conservation needs like EDGE program for example.

Corrected, we added the EDGE information

Lines 59-66: Among other threats Authors should mention the risk of Bd introduction which could lead to a possible (and unpredictable) mass mortality in the ponds.

  Corrected, we have modified it as suggested.

Lines 60-63: I would move this sentence as the first sentence of this paragraph and move the actual first sentence after it in order to first consider all the possible threats and then focus on invasive species.

Corrected, we have modified it as suggested.

Lines 69-70: can we consider this subspecies as an Evolutionary Significant Unit (ESU) in need of conservation?

Corrected, thank you for your precious comment. We added this information at the end of the Discussion section because it reinforces our points about the urgency of a management plan and conservation need (L 594-597).

Lines 83-85: Added to the fact that global change will increase the probability of extreme climatic events. Maybe Authors could add on sentence about it?

Corrected, we added a reference indicating a similar issue found for another Italian species.

Lines 109-110: Authors could indicate the total number or recognized subspecies in the alpestris species?

Corrected, we added this information when discussing the relevance of the inexpectata clade, in L 592-594.

Line 174: it could be useful the remind here the range of the breeding season.

Corrected.

Lines 175-176: what kind of weather change can prevent sampling?

We found that, for the features of the study sites, sudden thick fog hindered the field sampling, also considering that the two new sites are not easily accessible. We prefer not to add this information in the text, as it sounds like a technical guideline than a “regular” paper method.

Line 186: Please indicate if the operators wear gloves for this manipulation?

 Corrected.

Line 239: Formula: it could be helpful to remind the reader the meaning of Mi, L0, Li and bSMA.

Corrected.

Line 337: What do you mean by “evidence of predation”? Can you describe these evidences?

 Corrected, we better specified what we meant.

Line 347: “I. a. inexpectata”: in line 345 you use the common name; I think you should be consistent with the use of scientific or common name in the manuscript.

Corrected, we have modified it as suggested.

 Lines 372-373: have you tested the effect of presence or absence of introduced fished on the mean SMI of the newts? 

As we could sample and measure newts from four sites, of which one had fish presence (TRIF), we could not give statistical power to any analysis. Anyway, no SMI did not differ statistically between fish-invaded and fishless sites. See also L 552-556.

 Line 475: general comment for the intro of the “Threats and conservation status” paragraph: it would be interesting to discuss the interest of focusing conservation effort not only at the species level but also at the subspecies level or even at the population or morphotype level that could be considered as Evolutionary Significant Unit who deserve conservation effort.

Corrected, we added a specific sentence about it in L 594-597.

Line 519-520: does quantitative eDNA analysis could help to confirm the absence and reinforce population the density assessment?

 Corrected, we added this hint in the following lines.

Figure 1: Depending on what authors want to illustrate on this map I would suggestion the following changes:

         1) if Authors want to present the localization of I. a. inexpecta among the other subspecies of Alpine newts, the map should be completed by extending I. a. alpestris to the North and completing the map in the Eastern part by adding I. a. montenegrina and a part of I. a. carpathica distribution too.

2) if Authors want to show only the two Italian "endemic" subspecies the nominal subspecies should be deleted from the map.

Corrected, we changed the Figure 1 following the second suggestion you made.

Figure 2: Please, indicate (a) on the left figure and (b) on the right one.

Corrected.

Figure 3: I would suggest to remove the LG site from the figure 3b as this site is clearly apart from the others (visible in figure 2) and do not appear in figure 3a.

The LG site is actually present in Figure 3a, and it was possibly unnoticed because of the mountain shading background and the graphics in general. For the sake of completeness, and also considering that the PG site (which is also far from the ‘core sites’ we are analyzing, but indeed deserving attention), we prefer to keep all sites, simply indicating with the arrows that these two sites are not in the frame. Indeed, to avoid readers’ confusion, we highlighted with a white shade the two sites’ labels.

Figure 4: Would it be possible to indicate on the graphs the months where larvae have been observed?

Corrected, we added this information in Figure 4, indicating the presence of larvae through arrows.

Table 1: 23 animals observed in Laghi di Fagnano: does this number refers to animals observed in 2018 and 2022?

As outlined in the Table 1 caption, the number reported is the “Maximum number of adults captured or observed is based on information retrieved from the available literature or collected ante 2022 in the DiBEST database”; thus, 23 is the maximum number ever found in the years before 2022. We also better clarified this issue by changing the data in the table.

Reviewer 3 Report

Dear Editor,

The manuscript “Survived the glaciations, will they survive the fish? Allochthonous ichthyofauna and endemic newts: a road map for a conservation strategy” by I. Bernabò with co-authors is devoted to an important problem of the Calabrian Alpine newt populations conservation in the Southern Italy. This endangered species is an endemic taxon on the brink of extinction, deserving of urgent conservation actions. The paper presents data on the population density of the newt, the impact of invasive fish species on the decline of newt populations, which is of great importance and concern to the scientific community. The work addresses important issues such as an updated distribution of the Calabrian Alpine newt, species occurrence and habitat characteristics, threats and conservation status, as well as the conservation measures and active management plan. Convincing evidence of the negative effects of allochthonous fish species on the abundance decrease of the newt and its extinction was presented, which is a huge recent problem for many amphibian species around the world. Further population monitoring and demographic estimates in the breeding sites appears essential to maintain population stability.

In general, I think the scope of the manuscript “Survived the glaciations, will they survive the fish? Allochthonous ichthyofauna and endemic newts: a road map for a conservation strategy” and its findings fit with the aims of Animals Journal and should be of interest to its readers. The topic itself is very attractive and their work responds interesting questions to improve our understanding about amphibian conservation biology. The work is well structured, written in a scientific style, competently. The article gives the impression of a completed well-described study and can be accepted in its original form. My conclusion is the full acceptance of the article.

Best regards,

Author Response

Dear Reviewer, we really appreciate your comments and your positive feedback on our manuscript.